# A Polylactide-Based Micellar Adjuvant Improves the Intensity and Quality of Immune Response

**DOI:** 10.3390/pharmaceutics14010107

**Published:** 2022-01-03

**Authors:** Myriam Lamrayah, Capucine Phelip, Céline Coiffier, Céline Lacroix, Thibaut Willemin, Thomas Trimaille, Bernard Verrier

**Affiliations:** 1Laboratoire de Biologie Tissulaire et Ingénierie Thérapeutique (LBTI), Institut de Biologie et Chimie des Protéines (IBCP), CNRS UMR 5305, Université Lyon 1, Université de Lyon, 69367 Lyon, France; capucinephelip@gmail.com (C.P.); celine.coiffier@ibcp.fr (C.C.); celine.fish@free.fr (C.L.); thibaut.willemin@gmail.com (T.W.); bernard.verrier@ibcp.fr (B.V.); 2Laboratoire Ingénierie des Matériaux Polymères (IMP), CNRS UMR 5223, Université Lyon 1, Université de Lyon, 69622 Villeurbanne, France

**Keywords:** micelles, vaccine delivery, immunization, adjuvants, nanoparticles, biodistribution, p24

## Abstract

Micelles from amphiphilic polylactide-block-poly(*N*-acryloxysuccinimide-*co*-*N*-vinylpyrrolidone) (PLA-b-P(NAS-*co*-NVP)) block copolymers of 105 nm in size were characterized and evaluated in a vaccine context. The micelles were non-toxic in vitro (both in dendritic cells and HeLa cells). In vitro fluorescence experiments combined with in vivo fluorescence tomography imaging, through micelle loading with the DiR near infrared probe, suggested an efficient uptake of the micelles by the immune cells. The antigenic protein p24 of the HIV-1 was successfully coupled on the micelles using the reactive *N*-succinimidyl ester groups on the micelle corona, as shown by SDS-PAGE analyses. The antigenicity of the coupled antigen was preserved and even improved, as assessed by the immuno-enzymatic (ELISA) test. Then, the performances of the micelles in immunization were investigated and compared to different p24-coated PLA nanoparticles, as well as Alum and MF59 gold standards, following a standardized HIV-1 immunization protocol in mice. The humoral response intensity (IgG titers) was substantially similar between the PLA micelles and all other adjuvants over an extended time range (one year). More interestingly, this immune response induced by PLA micelles was qualitatively higher than the gold standards and PLA nanoparticles analogs, expressed through an increasing avidity index over time (>60% at day 365). Taken together, these results demonstrate the potential of such small-sized micellar systems for vaccine delivery.

## 1. Introduction

Much of the current effort related to vaccines is focused on developing vaccines based on molecularly defined antigens, which present excellent safety profiles over traditional vaccines based on live-attenuated or inactivated viruses. While antigen encoding nucleic acids (i.e., mRNA) have proven tremendous potential in the context of the SARS-CoV-2 pandemic (with the first mRNA-based vaccines from Pfizer/BioNTech and Moderna approved for humans) [1,2], some issues need to be addressed, such as the high dose of mRNA required or the need for drastic storage conditions linked to mRNA instability and sparse knowledge on the long-term intensity/quality of the immune response. In this regard, protein antigen-based vaccines remain a robust and attractive technology, as highlighted by the forthcoming approval of such a vaccine against COVID-19 from Novavax (NVX-CoV2373) whose strategy is based on self-assembled amphiphilic micelles. These amphiphilic copolymer micelles are gaining increasing attention in vaccine delivery due to their small size and versatility favorable for chemical design [3].

Such “subunit” vaccines require the use of adjuvants since the free antigen induces a poor inherent immunogenicity [4]. To date, there are still very few adjuvants employed in licensed human vaccines (e.g., Alum, MF59, AS04). Besides their mechanisms of action that are still under scrutiny, their safety could remain questionable [5,6], resulting in growing negative perceptions from populations, particularly regarding infant vaccination. Therefore, intensive research efforts have been devoted to nano/micro systems as potential adjuvants, such as liposomes [7], emulsions [8], dendrimers [9] and polymeric particles [10]. In particular, polylactide (PLA)-based nanoparticles (NP) with either encapsulated (into the core) [11] or adsorbed (at the surface) [12,13] antigens represent safe adjuvants able to compete with Alum or MF59 standards. PLA is a biocompatible, biodegradable, Food and Drug Administration (FDA)-approved polymer and several drug delivery formulations, biomedical devices and tissue engineering scaffolds derived from this polymer have been marketed over decades [14,15].

On the other hand, micelles from amphiphilic copolymers, which have been highly explored in relation to drug (e.g., anti-cancer) delivery [16], have recently emerged as highly relevant adjuvants for vaccines. They indeed present interesting features over non-micellar adjuvant systems [17,18]: (i) they are easy to prepare with a solid batch-to-batch reproducibility, (ii) due to their small size (generally <100 nm), they facilitate antigen delivery to dendritic cells (DCs) in draining lymph nodes. These micelles do not limit their association with DCs from the injection site, but they are capable of targeting them by traveling through the lymphatic system directly to the lymph nodes, where DCs are in a greater concentration compared to the periphery, (iii) they can easily display suitable surface properties (surface charge, reactive moieties) through the appropriate choice of biocompatible hydrophilic segments of the micelle corona [19,20,21]. Still, a sufficiently low critical micellar concentration (CMC) is desirable to avoid the premature disassembly of the micelles via dilution following the injection. Different micellar systems have been developed for vaccine delivery over the last years, but many of them are derived from non-degradable [19,22,23,24,25], cationic and potentially toxic compounds [26,27,28,29]; thus, they are often not clinically relevant. In addition, many developed micellar systems for vaccines contain poly(ethylene glycol) (PEG), a widely used FDA-approved pharmaceutical excipient as hydrophilic corona [21,30,31,32]. However, the involvement of such a polymer in hypersensitive reactions (production of anti-PEG antibodies) is reported to be more and more critical [33,34]. Besides this issue, PEG, as a polyether, offers poor flexibility for the coupling of biologic entities by chemical derivatization (possible only at the PEG chain end). Thus, there is still a need to develop safe, robust and flexible micellar carriers to address vaccine challenges.

In this context, we have previously developed micelles from an amphiphilic block copolymer based on FDA-approved PLA and hydrophilic poly(*N*-vinylpyrrolidone) (PNVP), which incorporate *N*-succinimidyl (NS) activated esters along the chain, allowing the coupling of antigens and/or immunomodulating compounds in a high density [35,36] and showed their potential as adjuvants in vitro [37]. When coupled to proteins, they preserved a relatively low CMC (around 10 µg/mL) [36]. In this paper, we explore the potential of the block copolymer micelles in depth and characterize them through an evaluation of cytotoxicity, in vitro cell uptake and in vivo biodistribution. Their performances in immunization are finally investigated and compared to Alum and MF59 gold standards as well as PLA NP through the intensity (IgG titers) and quality (avidity index) of the immune response.

## 2. Materials and Methods

### 2.1. Materials

DiR ethanol XenoLight Fluorescent Dye (DiIC18(7), 1,1′-dioctadecyltetramethyl indotricarbocyanine Iodide) was purchased from ThermoFisher Scientific (Waltham, MA, USA). HIV-1 gag p24 antigen was purchased from PX’Therapeutics (Grenoble, France) and endotoxins were removed as previously described [38]. The purity of p24 was higher than 97% with an endotoxin content lower than 5 EU/mg of p24 protein, as determined using the Quantitative Chromogenic Limulus Amebocyte Lysate (LAL) kit (BioWhittaker, Walkersville, Verviers, Belgium). The poly(d,l-lactide)-b-poly*N*-acryloxysuccinimide-*co*-*N*-vinylpyrrolidone) (PLA-b-P(NAS-*co*-NVP)) block copolymer (17,000 and 14,000 g/mol for PLA and P(NAS-*co*-NVP) respectively, Đ = 1.6) was synthesized as previously described [39].

### 2.2. Preparation of PLA-Based Micelles

Micelles were prepared as previously reported [37]. In brief, 5 mL of a copolymer solution (10 mg/mL) in acetonitrile (Carlo Erba Reagents, Peypin, France) was added to 10 mL of milli-Q water under agitation (200 rpm), allowing the formation of micelles. Acetonitrile and a part of water were removed by evaporation under reduced pressure using a Rotavapor R-300 (Buchi, Villebon sur Yvette, France). The micelle concentration was determined by measuring the solid content, after heating a known volume of the micellar solution to a constant weight in an oven at 70 °C for 24 h. The micelle solution (500 µL, 5 mg/mL) was further incubated with DiR (0.26 wt%/micelle, 1 µL of a 6.5 mg/mL of DiR solution in ethanol) overnight, protected from the light. DiR encapsulation was directly evidenced on the micellar solution by fluorescence (750 nm/780 nm) and visible spectrometry using a Tecan i-control Infinite M1000 (Tecan, Männedorf, Switzerland). DiR loading efficiency was determined by fluorescence intensity on a micellar solution after dialysis (3500 Da cut-off) and 10-fold dilution in acetonitrile, using a calibration curve established under the same conditions. The micelles were diluted 2-fold in Phosphate Buffer Saline (PBS) at pH 7.4 (Gibco, Dublin, Ireland) and incubated overnight before further biological studies.

### 2.3. Physico-Chemical Characterization

The hydrodynamic diameter, polydispersity index (PdI) and zeta potential of the micelles and formulated micelles were determined by Dynamic Light Scattering (DLS) analysis with the Zetasizer NanoZEN S600 device (Malvern Instruments, Malvern, UK) at 25 °C. The samples were prepared by 1/100 dilution in a solution of NaCl 1 mM filtered over 0.22 µm. The data were obtained by Zetasizer Software 7.11 (Malvern Instruments, Malvern, UK). The values were the mean of four measurements.

### 2.4. Cell Culture Protocol

The cell lines used were HeLa (ATCC^®^ CCL-2™) and a murine DCs line (DC 2.4 [40], an immortalized murine bone marrow derived DCs line), and were grown according to the typical culture procedure detailed here. The cell culture medium for DCs was composed of RPMI-1640, heat-inactivated Fetal Bovine Serum (FBS) (10%), 2-mercaptoethanol (50 µM) and HEPES buffer solution (10 mM). HeLa cells were cultured in DMEM containing 10% heat-inactivated FBS (all reagents were purchased from Gibco, Ireland and Life Technologies, Carlsbad, CA, USA). For both cell lines, after aspirating the old culture medium, the cells that adhered to the bottom of the flask T75 were washed twice with 10 mL of PBS. Then, 1 mL of trypsin solution (0.25% trypsin-EDTA) was added to the cells and let for 3–10 min at 37 °C. Then, the trypsin solution containing the cells was mixed with 9 mL of fresh complete culture medium. After centrifugation (5 min at 300× *g*), the appropriate amount of cells was resuspended in 13 mL of fresh culture medium in a new T75 flask, and cultured in a 37 °C incubator (Heracell 150i, ThermoFisher Scientific, USA) under 5% CO_2_ and 95% humidity. Cells were used with a low passage number (less than 10).

### 2.5. In Vitro Fluorescence and Cytotoxicity Studies

HeLa and DC 2.4 cells were seeded at a density of 20,000 cells/well into 96-well plates for 6, 24, 48 and 72 h. HeLa and DC 2.4 cells were incubated with 90 µL of serum-free medium containing the DiR-loaded or free micelles (22.5 µg, 2.25 µg and 225 ng of micelles/well). After incubation, the medium was aspirated and the wells were washed three times with PBS. Cells were trypsinized during 10 min at 37 °C. Then, the trypsin solution containing the cells was mixed with 120 μL of fresh medium. The cells were transferred to a 96-well V-bottom plate for centrifugation for 6 min at 1100× *g*. The pellets were resuspended in 100 μL of PBS and transferred to a 96-well black plate. DiR fluorescence intensity was measured with Tecan i-control Infinite M1000 (750 nm/780 nm).

At each determined time point, the cytotoxicity of the micelles was evaluated by Presto Blue Assay (ThermoFisher Scientific, USA) according to the manufacturer’s instructions. Briefly, 10 μL of Presto Blue Reagent was added and the plates were incubated for 15 min at 37 °C. The PLA micelles and the free fluorophore were used as controls (at 22.5 µg of micelles/well and at 0.06 µg of DiR/well, respectively, corresponding to the quantities of the 1/10 micelles-DiR condition). Fluorescence was detected on Tecan i-control Infinite M1000 (560 nm/590 nm). Fluorescence was determined as the mean of three replicates and four independent experiments.

### 2.6. In Vivo Biodistribution Studies

SKH1 female mice (*n* = 5 per group) were bred at Charles River Laboratories (L’Arbresle, France) and housed at the AniCan animal facility of the CRCL (Cancer Research Center of Lyon, Lyon, France). The experiments were approved by the relevant local ethics committee (CECCAPP_CLB_2017_006) and were conducted according to the rules for the care and use of laboratory animals.

Sixteen-week-old mice received a single subcutaneous (SC) injection targeting the left inguinal lymph node or a single IV injection of 0.96 µM DiR/mouse loaded into micelles (injection volume: 100 µL). Fluorescence intensity from the whole body and from the injection site for the SC group were recorded at different time points (5 min, 6 h, 1, 2, 3, 4, 7, 14, 21, 28 and 35 days after injection) using the FMT4000 fluorescence tomography imaging system (Perkin Elmer, Waltham, MA, USA). A group control receiving the free probe was performed (*n* = 3 mice), and the fluorescence was recorded at five time points (5 min, 6 h, 24 h, 48 h and 7 days after injection). For each imaging time point, mice were anesthetized under 3% isoflurane and positioned in the FMT system imaging chamber. The filter set was chosen depending on the fluorophore parameters (channel 745 nm/770 nm). The collected fluorescence intensities were reconstructed using the TrueQuant software (v4.0, Perkin Elmer, USA) for the quantification of a three-dimensional fluorescence signal and the acquired images were analyzed by drawing regions of interest (ROI). The total amount of fluorescence and at the injection site (in pmoles) per ROI were generated for all studies, and then the ratio of local fluorescence compared to the whole body was calculated as injection site residual fluorescence (%) = [amount of fluorescence at injection site/amount of fluorescence in whole body] × 100.

### 2.7. p24 Antigen Immobilization

The immobilization of the p24 protein on the micelles was performed by adding 300 μL of micelle solution (5 mg/mL) to the same volume of p24 in PBS (pH 7.4) at a concentration of 0.6 mg/mL (final concentrations in the coupling medium: 2.5 mg/mL for the micelles and 0.3 mg/mL for the protein p24). The samples were allowed to stir overnight. The coupling was assessed by SDS-PAGE (sodium dodecyl sulfate polyacrylamide gel electrophoresis) as previously described [36]. In brief, the micelle aqueous solutions were mixed with the carrier buffer (Laemmli Sample 5× Buffer: 300 mM Tris-Cl pH 6.8, 10% SDS, 40% glycerol, 10 mM dithiothreitol, 0.05% bromophenol blue) (micelle/carrier buffer: 4/1 *v*/*v*). The migration was carried out at 100 V for 10 min and at 200 V for 40 min. Both gels (separation and concentration) were used for revelation. The gels were further stained with Coomassie blue staining solution (ThermoFisher Scientific, USA).

### 2.8. Antigenicity of Micelle-Immobilized p24

Ninety-six-well plates (Nunc MaxiSorp, ThermoFisher Scientific, USA) were coated overnight at room temperature (RT) with 100 μL of p24 immobilized on micelles or free p24 at a solution concentration of 10 μg/mL in PBS. After removing the solution, the plates were blocked for 1 h at 37 °C with 250 μL of PBS containing 10% horse serum and washed 3 times with PBS containing 0.05% Tween 20 (PBS-T) using a ThermoScientific autoplate washer according to a program in three steps: (i) 3 washing cycles with 300 μL PBS-T, (ii) soaking during 20 s, (iii) aspiration (normal mode, aspiration height: 2.6 mm and high speed). The same procedure was applied for the subsequent washings. Then, 100 μL/well of biotinylated rabbit anti-p24 polyclonal antibody (bioMérieux, Marcy l’Etoile, France) in PBS-T containing 10% horse serum was added (5-fold serial dilutions) and the plate incubated for 1 h at 37 °C. Following washes with PBS-T, the plates were reacted with peroxidase-conjugated streptavidin at a 1:20,000 dilution (from a 1 mg/mL solution, Jackson Immunoresearch, West Grove, PA, USA) in PBS-T containing 10% horse serum for 30 min at 37 °C. Plates were washed with PBS-T, revealed using 100 μL/well of tetramethylbenzidine (TMB) substrate (OptEIA™, BD Biosciences, Franklin Lakes, NJ, USA) and incubated for 30 min, protected from the light. The reaction was stopped with 100 μL of 1N sulfuric acid and the optical density at 450 nm (OD450) with a correction at 620 nm (OD620) was measured using a microplate reader (Multiskan FC, ThermoFisher Scientific, USA).

### 2.9. Immunization Protocol

CB6F1/Crl female mice (*n* = 5 per group) were bred at Charles River Laboratories (L’Arbresle, France) and housed at the Plateau de Biologie Expérimentale de la Souris (PBES, ENS Lyon, France). The experiments were approved by the relevant local ethics committee (CECCAPP_ENS_2014_040) and were conducted according to the rules for the care and use of laboratory animals.

Six-week old mice were immunized by SC injection (targeting the left inguinal lymph node) at day 0, 21 and 42 (following the ADITEC -Advanced Immunization Technologies- consortium protocol for harmonization which recommend a prime-boost-boost protocol, one injection each three weeks), with a volume of 100 µL of various formulations containing 5 µg of p24 (p24 concentration of 50 µg/mL): micelles-p24, Alhydrogel^®^ (2% aluminium hydroxide gel, Invivogen, San Diego, CA, USA) mixed with p24 according manufacturer’s instructions, AddaVax™ squalene-based oil-in-water (Invivogen, USA) mixed with p24 according manufacturer’s instructions, and p24 coated PLA NP with mean sizes of 160, 180 and 200 nm, prepared as previously reported [41]. Mice were bled via the retro-orbital vein before and regularly after immunizations (sampled volume: 100 µL). The samples were heated for 30 min at 37 °C, then centrifuged twice at 16,000× *g* for 10 min and supernatants were stored at −20 °C for further analyses.

### 2.10. Antibody Titers

Sera were tested for the presence of p24-specific IgG by enzyme-linked immunosorbent assay (ELISA) at 0, 21, 42, 60, 106, 190, 272 and 365 days after immunization. Ninety-six-well Nunc maxisorp plates (ThermoFisher Scientific, USA) were coated with 100 μL of 1 μg/mL of p24 protein overnight at RT. The p24 excess was eliminated and plates were blocked with 200 μL of 10% non-fat dry milk in PBS for 1 h at 37 °C to prevent non-specific binding of the antibodies (Abs). Plates were washed 3 times with PBS-T using a Thermo Scientific autoplate washer. Serum samples from immunized mice at the indicated time points were serially diluted in Dulbecco’s PBS (D-PBS) containing 1% of BSA. Then, 100 μL of each sample in duplicate were incubated on blocked plates for 1 h at 37 °C. After washing with PBS-T (3 times), wells were then incubated 1 h at 37 °C with anti-mouse IgG-HRP conjugate (1:10,000) from Southern Biotech. Plates were washed again, revealed using 100 μL per well of TMB substrate (OptEIA™, BD Biosciences, USA) and stopped using 1N sulfuric acid. The OD450 and OD620 were measured using a microplate reader (Multiskan FC, ThermoFisher Scientific, Waltham, MA, USA).

### 2.11. Antibody Avidity

The avidity of induced p24-specific IgG was determined by the antibody–antigen binding resistance to the action of a selected detergent, here urea. ELISA were performed as described above except dedicated plates were subject, after serum incubation, to washings with either PBS-T containing or not 8 M urea (3 times). The avidity index (in percent) was calculated as the ratio: OD450 of urea-treated samples/OD450 of PBS-T-treated samples × 100. Antisera with index values exceeding 50% were ascribed a high avidity, those with index values of 30 to 50% were ascribed intermediate avidity, and those below 30% were ascribed a low avidity.

### 2.12. Statistical Analysis

Statistical analysis was performed using GraphPad Prism Version 9.0 software. Normality or lognormality of samples was tested using d’Agostino and Pearson or Shapiro-Wilk omnibus normality test. For cell viability assay and p24 coating efficiency, a two-way ANOVA coupled with Bonferroni multiple comparison post hoc analysis was carried out. On populations that failed the normality assay (in vitro DiR fluorescence, specific anti-p24 IgG titers and avidity indexes), the data are presented with median and the non-parametric equivalent Mann-Whitney U test for single comparison or Kruskal-Wallis test with Dunn’s post hoc analysis for multiple comparison was used. For the matched body fluorescence intensities, the non-parametric Friedman test followed with Dunn’s multiple comparison was used. For all, the significance level of statistic comparison is indicated in figures legend.

## 3. Results

### 3.1. Micelle Preparation and Fluorophore Loading

The micelles of PLA-b-P(NAS-*co*-NVP) (PLA micelles) were prepared by the common solvent method (nanoprecipitation) as previously reported [37]. The micelle solution was incubated with 0.26 wt% of a DiR probe. Encapsulation of the probe through micellar solubilization was clearly evidenced by fluorescence (Figure 1a) and visible (Figure 1b) spectrometry. The DiR loading efficiency determined through the dialysis method was determined to be 98% (*w*/*w*). The micelles were then incubated in PBS pH 7.4 (two-fold dilution) for further in vitro/in vivo studies. Final PLA micelles exhibited a mean size of 107 nm (PdI = 0.1) and a zeta potential of −32 mV. DiR loading did not significantly alter the micelle size and surface charge when compared to blank micelles. Furthermore, all the formulations were physically stable for at least one week (Table 1).

### 3.2. In Vitro Cytotoxicity and DiR Fluorescence Studies

The cytotoxicity profiles of the DiR-loaded micelles and the unloaded micelles were assessed by a Presto Blue assay on both HeLa cells and murine DC 2.4. Firstly, a slight decrease in the number of viable cells was observed over time, when treated with the micelles (either with or without DiR), whereas the non-treated cells showed a normal development. However, micelle-treated cells recovered normal growth after 48 h for DC 2.4 and 72 h for HeLa cells (Figure 2a,b). This tendency was not significant after statistical analyses.

We further examined the fluorescence of the DiR upon incorporation in the cell membranes. Interestingly, the intensity of the fluorescence of the micelles-DiR was much higher in DC than in the HeLa cells, as compared to the DiR control without micelles, suggesting a better internalization of the probe in DCs due to the micellization. Furthermore, the fluorescence increase was delayed in the time for the micelles-DiR compared to the control (DiR) as a result of DiR encapsulation in the micelles (Figure 2c,d).

### 3.3. In Vivo Biodistribution Studies

DiR-loaded micelles were injected subcutaneously into mice and their trafficking was monitored by fluorescence tomography over time (Figure 3a). The amounts of DiR rapidly increased in the whole body and at the injection site until three days, before reaching a plateau (Figure 3b), with an injection site/whole body fluorescence ratio varying little over time (22 ± 6%, Figure 3c). The free fluorophore injected in the same conditions was used as a control, but no signal could be detected in the whole organism at the five time points, due to the instant elimination. The micelles thus favored a depot effect at the injection site and an extended circulation time compared to free form, since the vectorization promotes the biological uptake of molecules. Moreover, based on tomographic reconstructed images, the fluorescence tended to accumulate preferentially in the spleen after three days (anatomically verified with IMAIOS©, www.imaios.com, accessed on the 7 May 2021), suggesting a transport of micelles (at least partially) through the lymphatic system in a first phase. Then, we can presume an integration of micelles in DCs where a release of DiR occurred in the cell membranes, leading to membrane fluorescence similar to that observed in vitro (Figure 2c,d). Except the two main accumulation sites (the red area following the intensity scale which represents the injection site and spleen), the fluorescence was broadly disseminated at a lower intensity, according to the repartition of the extensive lymph node chain.

### 3.4. p24 Immobilization on Micelles

For further vaccine evaluation, the p24 protein antigen of HIV-1 (120 µg amount per mg of micelle) was immobilized on the micelles through the coupling of its amines (*N*-terminal and lysine) on the *N*-succinimidyl ester moieties along the micelle corona in PBS pH 7.4. The SDS-PAGE analysis showed a nearly complete coupling efficiency after 24 h, as almost no free p24 was detected (Figure 4a). The concentration gel was kept for analysis since the protein-loaded micelles were too big to diffuse through the gel and could be observed at the start. A deeper kinetic coupling study showed that most of the protein was coupled within 3 h, with a further slower coupling process until 24 h (Figure 4b). The final antigen-conjugated micelles had a diameter of 114 nm, which was slightly higher than the blank micelles under the same conditions (105 nm), consistent with the coating of the protein. The zeta potential was also slightly affected, with a value of −37 mV, as compared to −30 mV. This was consistent with the isoelectric point of the p24 (PI = 5.9), implying that the p24 has a negative global charge at a neutral pH, and thus conferring a still significant negative zeta potential to the micelles after protein coupling.

### 3.5. Antigenicity of Micelle-Immobilized p24

An immune-enzymatic test (ELISA) was used to evaluate the integrity of the coupled antigenic protein, namely its recognition by anti-p24 Abs (Figure 4c). Interestingly, recognition of the coupled p24 was superior to that of free p24, probably due to their better availability and exposure at the surface of the micelles. Thus, interestingly, the immobilization of p24 on the micelles did not only alter its antigenicity but tended to improve it.

### 3.6. Immune Responses

SC immunizations were performed in mice following the standardized protocol (Aditec consortium), and humoral responses induced by the micelles-p24 were assessed and compared to gold standards Alum-p24 (Alhydrogel-p24) and MF59-p24 (AddaVax-p24). They were also compared to PLA NP with immobilized p24, prepared in the laboratory (Table 2), which has been described as a potent and safe adjuvant system [38,40,41]. As shown in Figure 5a, anti-p24 IgG titers in serum obtained after immunization with micelles-p24 compared favorably with the titers observed with the abovementioned adjuvants. Regarding a comparison with structurally similar (i.e., PLA-based) NP-p24 systems, it should be mentioned that immunization with micelles-p24 induced IgG titers similar (at least not significantly different) to PLA NP-p24 of the closest size (160 nm), while the Abs titers for PLA-NP of higher sizes were weaker (particularly at days 21 and 42, see Figure 5b), suggesting a size effect on the immune response.

### 3.7. Avidity Index

The quality of the produced Abs, known to be essential for protective immunity, was investigated through a determination of the avidity index. Different trends of the Abs avidity index over time were observed depending on the formulation. Interestingly, the avidity of Abs from immunizations with micelles tended to increase until one year, while those obtained from gold standards Alum and MF59 tended to decrease or at least stabilize over time (Figure 6a). Again, this shows a quite similar avidity index profile for the micelles and the NP of the smallest size (160 nm). As shown in Figure 6b, after one year, only the Abs obtained from immunizations exclusively presented a high avidity index (>50%), i.e., all the animals of these two groups, while the other adjuvants induced Abs with lower avidity index (high index for only one or two animals per group).

## 4. Discussion

If most of the current innovative vaccine strategies are based on self-assemblies of protein, such as the ferritin complexes [43], only a few approaches use synthetic polymers due to technical hurdles and the presence of potential toxic products. Indeed, chemical incompatibility and immunologic interference are two key challenges to overcome when formulated antigens are mixed [44,45]. For example, most polymeric micelles designed for vaccines, recently reviewed by our group [17], rely on polymers that are non-degradable and/or have a cationic surface (generally to enable the electrostatic interactions with widely used model ovalbumin or nucleic acid based antigens), and raising potential issues of short- or long-term toxicity. We have previously assembled micelles made of amphiphilic block copolymers based on FDA-approved biodegradable PLA and PNVP. They offer high flexibility regarding the incorporation of antigens and/or immune modulators via a core encapsulation or a surface coupling by the functional groups of the PNVP-based corona, and showed promising results in vitro in a vaccine context [37,46]. In the present study, we encapsulated a hydrophobic near IR fluorophore termed DiR (able to bind to cell membranes and fluoresce), to obtain a deeper insight into the in vitro/in vivo behavior and trafficking of such micelles. First, we showed that the micelles (either with or without loaded fluorophore) did not induce any significant in vitro cytotoxicity on both HeLa and DCs. Interestingly, the micelles showed a better uptake by DCs compared to HeLa cells, and in a gradual manner (increasing fluorescence until 72 h). Similarly, whole body imaging experiments (fluorescence molecular tomography analyses) showed that, after SC injection in mice, fluorescence progressively increased over the course of time, reaching a plateau after 72 h, suggesting that the micelles allowed a gradual release of the DiR in the membranes. Regarding location, we found that, in addition to a depot at the injection site, the fluorescence has a tendency to accumulate in the spleen and draining lymph nodes. These observations thus suggest an efficient uptake by the antigen-presenting cells of the draining lymphatic system until they reach the main lymphoid organ at 72 h post injection. It is important to note that the same biodistribution experiment was performed following an intravenous (IV) injection. The results showed that fluorescence was also progressive, but it accumulated in the liver, and no fluorescence was detected anywhere else at a relevant intensity (Appendix A). It can be hypothesized that the serum proteins present in the blood prevent micelle uptake by circulating antigen-presenting cells. Moreover, when the above mentioned in vitro fluorescence experiments (Figure 2b) were performed in the presence of serum, no fluorescence was detected in the cells, indeed emphasizing that serum most probably prevents integration of the micelles in the cells.

From the perspective of vaccine evaluation, our PLA-based micelles were further coupled to the p24 antigenic protein of HIV-1 through the *N*-succinimidyl esters present on the NVP-based hydrophilic block corona of the micelles. The coupling was nearly quantitative, as shown by SDS-PAGE analysis, while the size and surface charge remained quite unaffected, close to the native micelles. Interestingly, the vectorized p24 presented a better antigenicity than the soluble form, which could explain an improved presentation to the Abs, thanks to its fixation on the hydrophilic block of the micelle corona deployed in the solution. When injected subcutaneously in mice, the micelles-p24 induced IgG Abs titers comparable to those of the gold standards Alum and MF59. Antigen-coated PLA NP were also evaluated as positive controls, because they are structurally similar to our micelles and have been reported as relevant candidates for vaccine delivery [13,41]. Interestingly, the micelles induced an IgG titers profile similar to that of the PLA NP of the closest size (160 nm), while PLA NP of higher size (180 and 200 nm) induced a lower Abs titer in the first phase (until day 60). Regarding the quality of the immune response, the micelles induced Abs with a high avidity compared to the gold standards and were again similar to that of 160 nm-PLA NP. One hypothetical explanation of this strong humoral response with high avidity could be related to the B Cell Receptor (BCR) engagement due to the biodistribution profile of micelles through the draining lymph system, favoring a BCR clustering, as it has been observed with other self-assembled micelles [47,48] and NP [49,50]. To date, to our knowledge, no micellar-based system has shown a performance that can compete with vaccine gold standards (alum and MF59) by means of both intensity and quality of humoral immune response. To support this affirmation, the micellar effect on the cellular immune response is needed and is currently being studied by the group.

It is noteworthy that the avidity tended to decrease when the size of the NP increased. These results strongly suggest a size-dependent response of the vector on the intensity and on the quality of the immune response, consistent with previous works showing a critical impact of size, particularly in the tiny range of 100–200 nm, for which the mechanism may switch from passive lymph drainage to active cell-mediated transport [51,52,53,54].

Nevertheless, factors other than size could be influential in the performances of this micelle system. Indeed, for example, the antigen density (multivalency) has been shown to be crucial in the context of HIV-1 vaccine based on Env proteins, as well as their covalent binding to the carrier [51,55]. Our micelle platform here enabled a p24 density of 120 µg/mg of micelle, which was quite important, with a covalent binding. These favorable features may also explain their performance in relation to vaccination. Even though a model antigen was used for the proof of concept here, these encouraging results should motivate further investigations of this micellar platform in the vaccine field.

## 5. Conclusions

We described the potential of small-sized PLA-based micelles in subunit vaccine context. Through near IR probe encapsulation, we showed in vitro and in vivo that these non-toxic micelles were prone for uptake by DCs and carried through both the antigen presenting cell migration and the lymphatic system. They were also versatile as they could be easily and highly surface modified with a dedicated antigen for vaccine delivery, preserving first but even most importantly improving its antigenicity. The intensity and quality of the humoral responses induced by the micelles in HIV-1 vaccination model were comparable and higher, respectively, to those obtained with the gold standards (Alum/MF59), showing the relevance of such micellar carriers in vaccine delivery. Further loading/coupling of immune modulators on this versatile micellar platform are ongoing, with a view to controlling the orientation of the immune responses and improve vaccine efficiency.

## Figures and Tables

**Figure 1 pharmaceutics-14-00107-f001:**
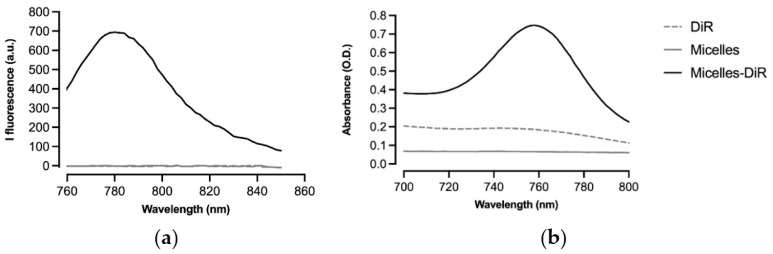
DiR loading in the micelles through micellar solubilization assessed by: (**a**) fluorescence (excitation wavelength: 750 nm); and (**b**) visible spectrometry. Thus, the formulation preserves the fluorescent capacities of the near infrared (IR) reagent.

**Figure 2 pharmaceutics-14-00107-f002:**
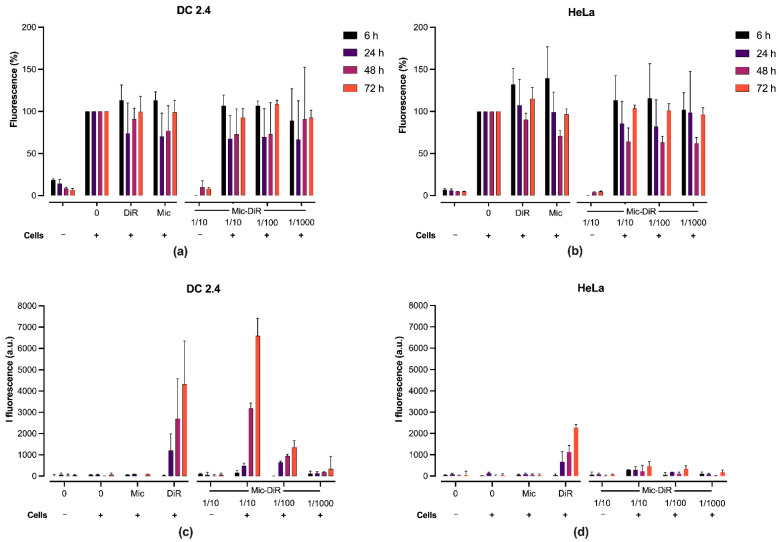
Cell viability was assessed by the Presto Blue test (560 nm/590 nm) at determined time points using (**a**) DC 2.4 and (**b**) HeLa cell lines. Fluorescence intensity of formulations using (**c**) DC 2.4 and (**d**) HeLa cell lines. Both were incubated with 90 µL of a serum-free medium containing the DiR-loaded or free micelles 10-, 100- and 1000-fold diluted (22.5 µg, 2.25 µg and 225 ng of micelles per well, respectively). The results are presented as the median (95% CI) of four independent experiments performed in triplicate. For figures (**a**,**b**), the results were transformed into percentages using the control (0 substance, with cells) as 100%. Furthermore, the normality of samples was validated using the d’Agostino and Pearson tests. A two-way ANOVA coupled with a Bonferroni multiple comparison post hoc analysis was carried out. Groups in the presence of cells showed no significant difference between each other at each study time. For figures (**c**,**d**), statistical significance between DiR and 1/10 Micelles-DiR groups was calculated using the two-tailed non-parametric Mann-Whitney U test. No significant difference was observed.

**Figure 3 pharmaceutics-14-00107-f003:**
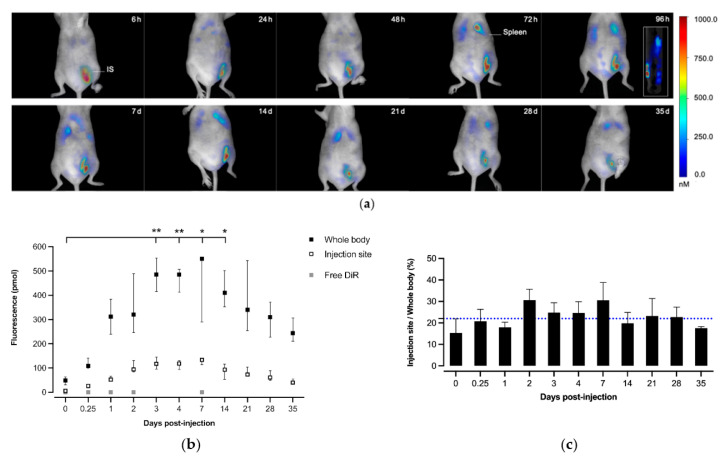
Distribution of DiR fluorescent micelles after their administration by SC route. (**a**) Fluorescence tomography imaging in the same mouse for 35 days (IS: injection site), with a sagittal view for a 96-h time point. The intensity scale of the fluorescence is in nmol/L of DiR. (**b**) Whole body and injection site fluorescence in function of post-injection time. The free DiR group was detected at five time points (5 min, 6 h, 24 h, 48 h and 7 days). The results are presented as the median (IQ). The intensity of the whole body fluorescence was compared each time against t0 using a non-parametric Friedman test followed by Dunn’s multiple comparison (*: *p* < 0.05; **: *p* < 0.001). A significant plateau was reached three days after injection, until 14 days. Nonsignificant differences are not represented. (**c**) The injection site to whole body fluorescence ratio in function of post-injection time with 22% of the mean at all times (blue dashed line), the results are presented as the mean (SD) (*n* = 5 mice).

**Figure 4 pharmaceutics-14-00107-f004:**
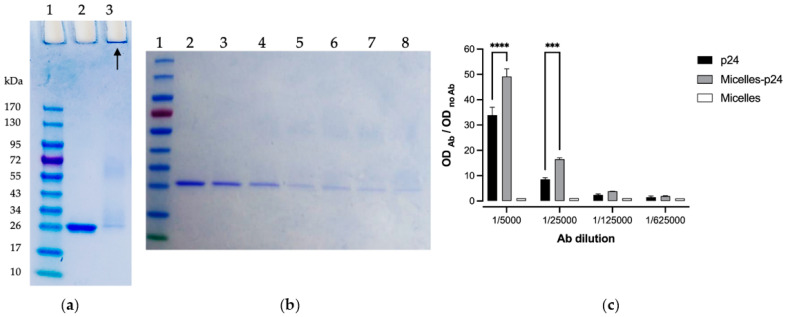
(**a**) SDS-PAGE analysis of p24 protein coupling on the micelles after 24 h, 1: Mass marker, 2: Free p24 ([p24] = 0.3 mg/mL), 3: Micelles conjugated to p24 ([Mic] = 2.5 mg/mL and [p24] = 0.3 mg/mL). (**b**) SDS-PAGE coupling analysis of the p24 protein ([p24] = 0.3 mg/mL) on micelles ([Mic] = 2.5 mg/mL) in the course of time: 1: mass marker, 2:0 h, 3:30 min, 4:1 h, 5:3 h, 6:4 h, 7:7 h, 8:24 h. (**c**) ELISA of free or copolymer micelle-immobilized p24. Coating of p24 (free or immobilized) with a concentration of 10 µg/mL. The results are presented as the mean (SD) of the experiment performed in duplicate. The analysis of normality was performed using a Shapiro-Wilk test. Differences between the groups were analyzed by a two-way ANOVA coupled with a Bonferroni multiple comparison post hoc test (***: *p* < 0.0005; ****: *p* < 0.0001). For a better understanding, only the differences between the free p24 and micelles-p24 have been shown. The recognition of p24 by ELISA was significantly higher in the micelles-p24 group compared to free p24.

**Figure 5 pharmaceutics-14-00107-f005:**
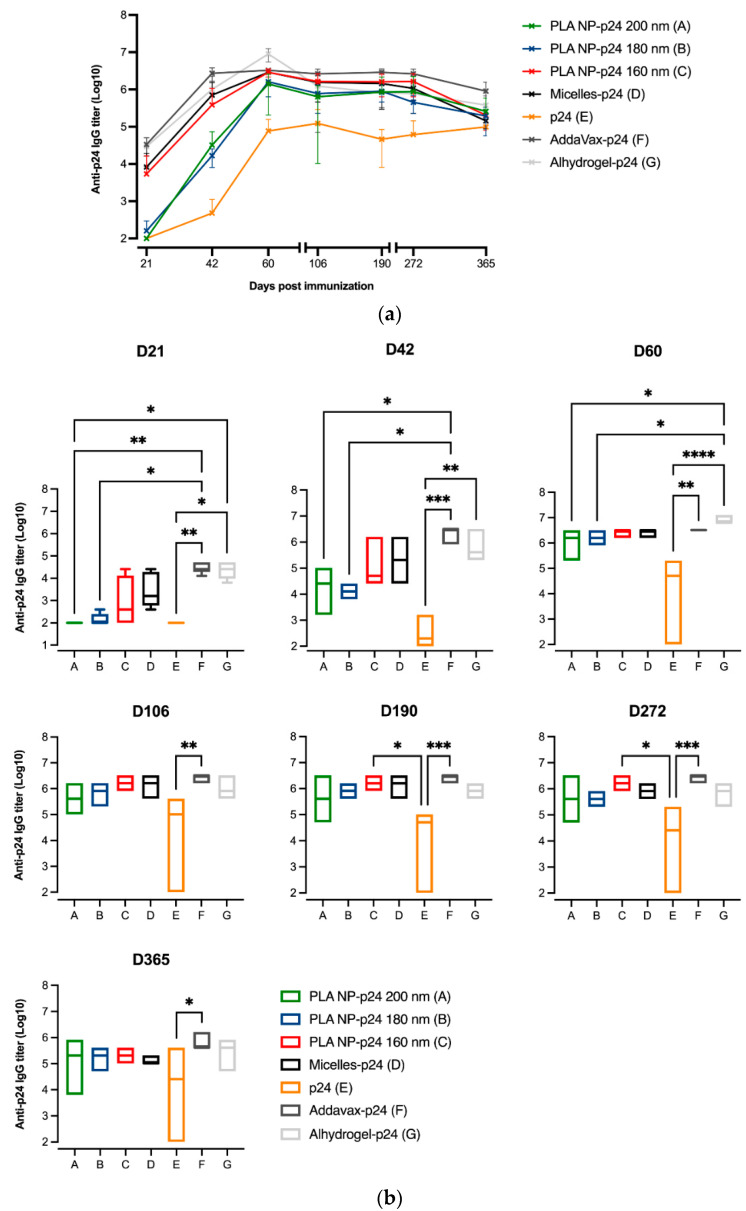
Specific anti-p24 IgG titers in the sera of mice immunized with micelles-p24, Alum (Alhydrogel)-p24 or MF59 (AddaVax)-p24 and PLA NP-p24. Animals (*n* = 5 per group) received administrations by the SC route of the different formulations containing 5 μg of p24 protein at days 0, 21 and 42 and IgG induction was monitored for one year; A: PLA NP-p24 200 nm; B: PLA NP-p24 180 nm; C: PLA NP-p24 160 nm; D: micelles-p24; E: p24; F: AddaVax-p24; G: Alhydrogel-p24. (**a**) Whole kinetics representation, results are presented as the mean (SD). (**b**) Data are represented with box and whiskers plots (min to max) and for each time point. Differences between the groups were analyzed on logarithmic data by Kruskal–Wallis with the post-hoc Dunn’s multiple comparisons test (*: *p* < 0.05; **: *p* < 0.01; ***: *p* < 0.001; ****: *p* < 0.0001).

**Figure 6 pharmaceutics-14-00107-f006:**
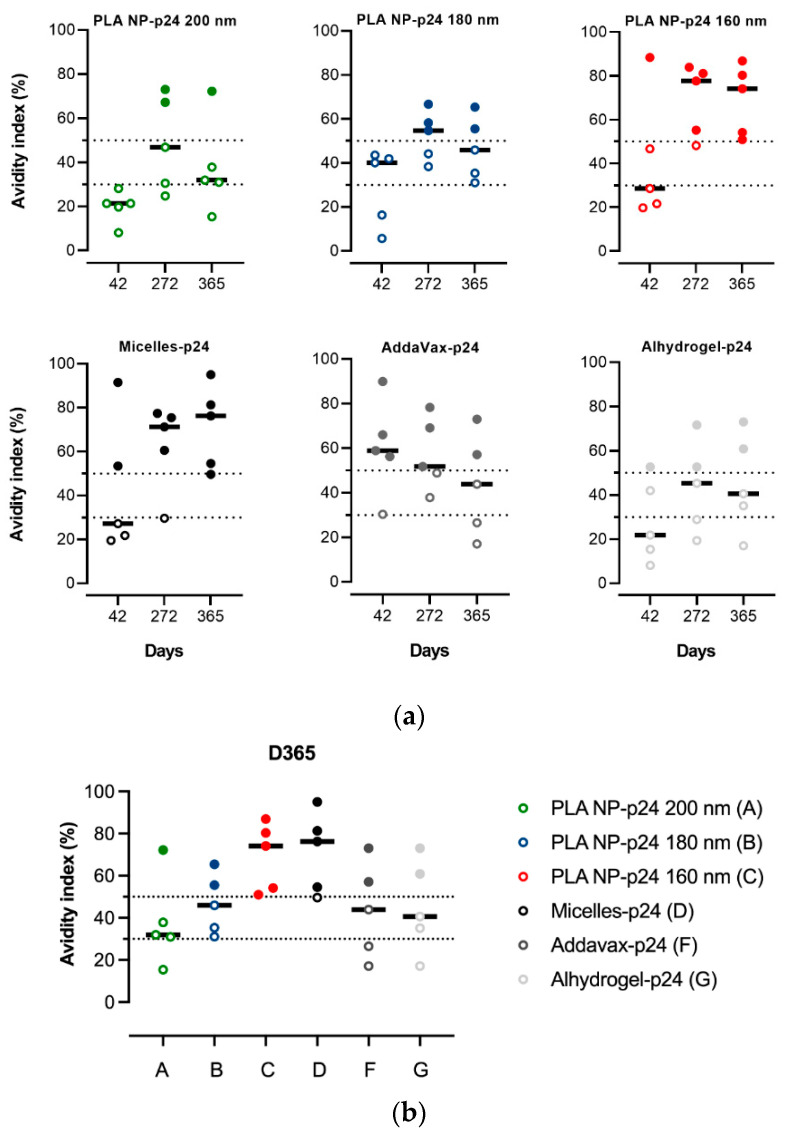
Avidity index of the anti-p24 IgG induced by immunization with the different formulations. Antisera with index values exceeding 50% were ascribed a high avidity, those with index values of 30 to 50% were ascribed intermediate avidity and those with index values of <30% were ascribed a low avidity [42]. (**a**) Avidity index tendency over time. Data points represent individual samples (*n* = 5 per group) with the line as the median. (**b**) Avidity index at day 365. Data are represented in dot plots with the line as the median. Differences between the groups were analyzed on non-parametric data by Kruskal-Wallis with post-hoc Dunn’s multiple comparisons test. No significant differences were observed. A: PLA NP-p24 200 nm; B: PLA NP-p24 180 nm; C: PLA NP-p24 160 nm; D: micelles-p24; F: AddaVax-p24; G: Alhydrogel-p24.

**Table 1 pharmaceutics-14-00107-t001:** Physico-chemical characteristics of the PLA micelles formulations with and without the near IR fluorophore DiR.

Formulation	Day Post Synthesis	Mean Size (nm)	PdI	Surface Charge (mV)
Micelles	0	102.0 (±0.9)	0.172 (±0.013)	−29.9 (±1.7)
Micelles	7	107.8 (±1.1)	0.091 (±0.042)	−35.5 (±0.5)
Micelles-DiR	0	107.0 (±1.7)	0.119 (±0.024)	−31.8 (±1.0)
Micelles-DiR	7	113.9 (±1.9)	0.162 (±0.006)	−36.5 (±0.5)

**Table 2 pharmaceutics-14-00107-t002:** Physico-chemical characteristics of the PLA-based micelles and NP formulations evaluated in immunization studies.

Formulation	p24 Adsorption Yield (%)	Mean Size (nm)	PdI	Surface Charge (mV)
PLA NP 160 nm	-	158.7 (±2.9)	0.068 (±0.005)	−54.4 (±0.5)
PLA NP-p24 160 nm	99.8	160.6 (±1.1)	0.073 (±0.023)	−59.2 (±2.1)
PLA NP 180 nm	-	181.7 (±6.3)	0.029 (±0.002)	−53.8 (±1.2)
PLA NP-p24 180 nm	89.2	184.8 (±2.0)	0.034 (±0.013)	−58.3 (±3.4)
PLA NP 200 nm	-	198.7 (±8.1)	0.044 (±0.027)	−59.2 (±2.3)
PLA NP-p24 200 nm	~100	197.9 (±1.1)	0.035 (±0.009)	−60.5(±0.9)
PLA-micelles	-	105.0 (±2.3)	0.063 (±0.021)	−29.9 (±5.8)
PLA-micelles-p24	~100	114.5 (±6.0)	0.040 (±0.026)	−37.2 (±0.8)

## Data Availability

Data are available from the authors upon reasonable request.

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
