# Peer review of "A Polylactide-Based Micellar Adjuvant Improves the Intensity and Quality of Immune Response"

_pharmaceutics, 2022, doi:10.3390/pharmaceutics14010107_

Round 1

Reviewer 1 Report

The manuscript entitled “Polylactide-based micellar adjuvant improves intensity and  quality of immune response” reported micelles formulated from PLA-b-P(NAS-co-NVP) block copolymer. In vitro fluorescence experiments combined with in vivo fluorescence tomography imaging based on the DiR-loaded micelles. Furthermore, antigenic protein p24 of the HIV-1 was successfully coupled on the micelles using the reactive N-succinimidyl ester groups on the micelle corona. The manuscript is organized generally well, and some preliminary results are given. The manuscript is recommended for publication consideration after following minor concerns.

  1. TEM analysis is suggested for the blank micelles of PLA-b-P(NAS-co-NVP), DiR-loaded micelle, as well as the p24-decorated micelles, respectively.
  2. The density of p24 in the micellar corona is recommended to be examined for further discussion.
  3. In Figure 3a, the mice images are suggested to be fully displayed for deliberate observations.
  4. For the background of this work, some recent literatures are suggested to be involved and referred: Chemical Engineering Journal 2021, 424, 130171; ACS Appl. Mater. Interfaces 2020, 12, 49489−49501; ACS Nano 2020, 14, 1919-1935.

Reviewer 2 Report

The authors provide results that demonstrate smaller miceller formulations can be used to encapsulate antigens resulting in Immune responses better than gold or conventional nanoparticles. 

Here are my comments:

My concern is figure 2. It is hard to understand how fluorescence has been used to calculate toxicity. It should either be converted to percent toxicity etc which seems high even in cells not treated with anything.

There is a figure under table two. Where does it belong? it seems free p24 and miceller p24 the titers are not that different. 

Why have they separated different sizes 160, 180, 200nm range is not that different. I think only one can be represented. or just pool all and provide a range. similarly Avidity is not that different. 

Is there any evidence of CD4 and CD8 responses?

Reviewer 3 Report

Authors have explored micelle-based formulation for vaccine development, which can be of interest and an alternative approach for vaccines. The in vivo data have shown good immune response, and the authors have tracked the data for one year, which is impressive. I have a few concerns before recommending the publication of this work.

1) in vitro cytotoxicity data should be discussed in % cell viability

2) For in-vitro internalization, a fluorescent/confocal microscopy image showing the internalization of particles is recommended. This can be much more informative to show whether the micelle goes inside the cell or binds to the cell membrane only?

3)Based upon in-vivo image, the author has claimed micelles are in the spleen and have reached the lymphatic system (in discussion). However, there is no confirmatory analysis for this. What is the verification process using IMAIOS? A histology-based experiment is needed to confirm the presence of micelles in the spleen and the authors claim in the discussion.

4) Figure 3. A control experiment with DiR only dye is required to show the difference of using a micelle-based delivery platform. Also, a mouse image at 0 h is required as a control.

5) Figure legend in Figures 5 and 6 is missing.

6) Authors have mentioned all the positive aspects in the discussion, looks like a perfect system. Are there any shortcomings as well of the micelle-based vaccine? 

Round 2

Reviewer 2 Report

Please include the Y axis in all panels of Figure 2 before final decision. I have no further comments.

Author Response

Figure 2 was modified following the recommendations of the reviewer 2 (addition of the y axis in figure 2B and 2D). 

Reviewer 3 Report

I recommend the manuscript for publication although confocal imaging showing internalization and histology-based confirmation of micelles in tissues would have added more value. 

Author Response

We would like to thank the advices of the reviewer 3, we keep in mind for the continuation of the project.